# Multipoint Wave Measurement in Tuned Liquid Damper Using Laser Doppler Vibrometer and Stepwise Rotating Galvanometer Scanner

**DOI:** 10.3390/s21248211

**Published:** 2021-12-08

**Authors:** Yoon-Soo Shin, Junhee Kim

**Affiliations:** Department of Architectural Engineering, Dankook University, Yongin 17058, Korea; shinys@dankook.ac.kr

**Keywords:** tuned liquid damper (TLD), laser Doppler vibrometer (LDV), galvanometer scanner, multipoint measurement

## Abstract

Liquid dampers, such as tuned liquid dampers (TLDs), are employed to improve serviceability by reducing wind-affected building vibrations. In order to maximize the vibration suppression efficiency of the liquid damper, the tuning frequency of the liquid damper should match the natural frequency of the building. Experimental evaluation of the tuning frequency of a liquid damper performed in a factory prior to installation in a building is a critical task to ensure correct performance, and for this, multipoint measurement of the TLD is required. In this study, a novel liquid level measurement system combining Laser Doppler Vibrometer (LDV) and a stepwise rotating galvanometer scanner was developed to observe liquid sloshing in TLD. The proposed system can measure the liquid level at multiple points simultaneously with a single laser point. In the experimental phase, the liquid damper’s natural frequency and mode shape are experimentally evaluated utilizing the developed system. The performance of the proposed system was verified by comparison with the video sensing system.

## 1. Introduction

Energy dissipation technology has been developed to install auxiliary mass dampers to improve the performance of flexible buildings sensitive to earthquake loads and wind [1,2]. A tuned mass damper (TMD) is a mechanical device that consists of a spring mass system. The secondary mass is a small fraction of the total mass of the primary structure and generates a reaction force induced by the oscillating motion of the mass [3,4]. Liquid dampers, on the other hand, are similar to TMDs, except for the use of oscillating liquid motion in gravitational fields [5,6]. The tuning frequency ratio, an important parameter in the design of a liquid damper, can be easily changed by the length of the liquid container [7,8,9]. Two different configurations of liquid dampers have been investigated: tuned liquid column damper (TLCD) and tuned liquid damper (TLD). These different configurations utilize energy dissipation wave braking/sloshing on free liquid surfaces and liquid motion of oscillating in U-shaped narrow tubes, respectively [10,11]. To accurately analyze liquid movement in the TLD, multipoint measurements are required to determine the sloshing mode of TLD. Unlike the TLCD, where the first oscillating mode is controlled, the first several modes are predominant in the TLD [12].

To date, capacitance wave gauges immersed into liquid have been used predominantly for various wave heights; however, several intrinsic shortcomings related to contact sensors have been continuously addressed: arduous installation, high price, limitation of measurement point, and loss of accuracy caused by interference from liquid medium (e.g., parasitic capacitance). While significant advances have been made in analytical studies of the practical design formulations/guidelines of liquid dampers [11,13,14] and their dynamic behavior [15,16,17], there are few studies on new measurement strategies [18].

Recently, there has been an increasing interest in noncontact sensing in structural monitoring and evaluation [19]. Typical types of contactless sensing include image processing using cameras and point data processing using lasers [20]. The precedent works of vision-based remote sensing are found in the literature [21,22,23,24,25]. Vision-based remote sensing has complemented some of the shortcomings of contact sensors, but there are still limitations in practical use, e.g., out-of-plane target sensing restrictions and only measuring liquids in transparent barrels. On the other hand, laser sensors such as LDV (laser Doppler vibrometer) and LiDAR (light detection and ranging) have high-performance precision and can measure in-plane motion; as such, they are being used in various fields of system measurement and analysis [26,27].

In this study, a high-precision LDV and galvanometer scanner-based sensing system was developed exclusively for the dynamic wave height measurement of the TLDs. A practical methodology for experimentally estimating the dynamic properties of TLD was compared based on the derived dynamic equation and measured experimental data [28]. A practical methodology for extracting the motion of the target liquid from the LDV data mixed with galvanometer motion noise is proposed based on experimental data. Finally, a series of experimental investigations was conducted to verify the proposed system and showcase the dynamic properties of liquids inside the TLD.

## 2. Considerations on the Proposed Scanning System

### 2.1. Natural Frequency of Liquid Motion in TLDs

A TLD generates a control force that reduces vibration of the building by the movement of liquid in a rectangular tank. The liquid in the TLD is a free surface motion based on shallow water theory.

Assuming a traveling wave propagation in a plane behaves as a trigonometric waveform, the vertical displacement of the surface elevation seen in Figure 1 is given as
(1)$$a\left(x,t\right)=A\cos\left(kx-\omega t\right)$$
where *A* is the wave amplitude and *k* and ω are the wavenumber and angular velocity, respectively. Similarly, propagation of velocities and pressure are expressed with trigonometric functions:(2)$$u\left(x,z,t\right)=U\left(z\right)\cos\left(kx-\omega t\right)$$
(3)$$w\left(x,z,t\right)=W\left(z\right)\sin\left(kx-\omega t\right)$$
(4)$$p\left(x,z,t\right)=p_{a}+\rho g\left(H-z\right)+P\left(z\right)\cos\left(kx-\omega t\right)$$
where $U\left(z\right)$, $W\left(z\right)$, and $P\left(z\right)$ are amplitudes of *u*, *w*, *p*, respectively, and depend on the depth, *z*. Considering the equation of motion of a nonviscous fluid and a mass conservation of fluid flow leads a differential equation of function of pressure at *z*, *P*(*z*):(5)$$\frac{d^{2}P\left(z\right)}{dz^{2}}=k^{2}P\left(z\right)$$

Of which the solution is:(6)$$P\left(z\right)=P_{1}e^{+kz}+P_{2}e^{-kz}$$
where $P_{1}$ and $P_{2}$ are two constants of integration to be determined from the following boundary conditions: (1) along the bottom $\left(z=0\right)$, there is no vertical velocity $\left(w=0\right)$. The boundary condition at the bottom demands $W\left(z\right)=0$ and dP/dz=0 at z=0. Thus $P_{1}$ = $P_{2}$ is obtained from Equation (6); (2) at the top of the fluid layer, along the deformable surface $\left(z=H+a\right)$, fluid particles move with the surface $\left(w=da/dt=\partial a/\partial t+u\partial a/\partial x+v\partial a/\partial y\right)$ under the atmospheric pressure. The condition leads to
(7)$$w\left(z=H\right)=\frac{\partial a}{\partial t}\;\mathrm{and}\;p\left(z=H\right)=p_{a}+\rho ga.$$

Equation (7) results in $W\left(z=H\right)=\omega A$ and $P\left(z=H\right)=\rho gA$. Substituting the conditions to Equation (6) yields
(8)$$kP_{1}e^{+kH}-kP_{1}e^{-kH}=\rho \omega^{2}A$$
(9)$$P_{1}e^{+kH}+P_{1}e^{-kH}=\rho gA$$

In the presence of a non-zero amplitude (A≠ 0), the ratio of Equations (8) and (9),
(10)$$k\;tanh\left(kH\right)=\frac{\omega^{2}}{g}$$

Imposes a relationship between the angular velocity ω and the wavenumber k in the traveling wave propagation:(11)$$\omega =\sqrt{gk\;\tan h\left(kH\right)}$$

Considering a rectangular basin of length L and with a flat bottom at depth H, an incident wave generates a reflected wave at the basin’s walls. The superposition of two traveling waves, i.e., incident and reflected waves, forms a standing wave termed mode. The long-wave approximation of the horizontal velocity is $u=\left(\omega A/kH\right)cos\left(kx-\omega t\right)$ [29] and the superposition of two such waves is warren as
(12)$$u=\frac{\omega A_{1}}{kH}\cos\left(kx-\omega t\right)-\frac{\omega A_{2}}{kH}\cos\left(kx+\omega t\right)$$

A boundary condition of the wall at x=0 is the horizontal velocity is $u\left(x=0\right)=0$. The condition implies $A_{1}=A_{2}$ and the horizontal velocity field can be then rewritten as
(13)$$u=2A\sqrt{\frac{g}{H}\sin\left(kx\right)\sin\left(\omega t\right)}$$

The other boundary condition of the wall at x=L is the horizontal velocity is $u\left(x=L\right)=0$. Using the condition, Equation (13) yields $sin\left(kL\right)=0$, which implies that the wavenumber k take the following values:(14)$$k=\frac{\pi }{L},\;\frac{2\pi }{L},\;\frac{3\pi }{L}\;\cdots$$

Using the condition of the wavenumbers of the standing wave, i.e., Equation (14), the modal frequencies (Figure 2) are theoretically derived from Equation (11):(15)$$f_{n}=\frac{1}{2\pi }\sqrt{\;\frac{n\pi g}{L}\tanh\left(\frac{n\pi H}{L}\right)},\;\;\left(n=1,\;2,\;3\;\ldots \right)$$

### 2.2. Working Principle of the LDV

The LDV measures a moving surface’s out-of-plane dynamic velocity and displacement with virtually no noise and nearly infinite sampling frequency. The LDV launches a sinusoidal laser beam at a moving target and measures dynamic velocity and displacement based on the interference of the incident laser beam with the beam reflected from the target. The reflected laser beam interferes with the reference beam, and as a result, the light intensity is measured by the photodetector installed on the scanning head of the LDV. The light intensity signal is converted from the decoder to displacement using the arctangent function. The operating principles are outlined in Figure 3.

The schematic wave of the modulated laser beam illustrated in Figure 4 can be expressed as follows:(16)$$u_{c}=u_{DC}+Acos\left(\omega_{c}t+\varphi_{m}\left(t\right)\right)$$
where $u_{c}$ is the light intensity detected by the photodetector and $u_{DC}$ is the DC component of the signal. In addition, $\omega_{c}$ is the frequency of the drive signal modulated and $\varphi_{m}$ is the phase angle that occurs when the reflection and reference laser beam are interrupted. After removing $u_{DC}$ from $u_{c}$ and lowpass filtering, two orthogonal signals $u_{i}$ and $u_{c}$ are extracted as
(17)$$u_{i}=\frac{A}{2}cos\varphi_{m}\left(t\right)\;\;\;\;\;\;u_{c}=\frac{A}{2}sin\varphi_{m}\left(t\right)$$

Phase angle is determined from the two signals as
(18)$$\varphi_{m}\left(t\right)=\arctan\left(\frac{u_{c}\left(t\right)}{u_{i}\left(t\right)}\right)$$

After obtaining $\varphi_{m}\left(t\right)$ from Equation (18), velocity and displacement are calculated by different LDV digital decoder processes [30]. The $\varphi_{m}\left(t\right)$ obtained from the arctangent method has a wrapped shape, because the output range of arctangent function is limited to $\left(-2\pi \;2\pi \right)$; therefore, an unwrapping process is required to reconstruct the phase angle $\varphi_{m}\left(t\right)$ and calculate dynamic displacement. On the other hand, the unwrapping process is unnecessary for velocity measurement, since the velocity is obtained by differentiation of $\varphi_{m}\left(t\right)$.

### 2.3. Proposed Scanning System

A galvanometer scanner equipped with a mirror at the end of a servo motor rotates at a constant speed when a signal is input to the servo motor. When used in conjunction with a laser, it is possible to change the scanning direction of the laser through the rotation of the mirror so that multiple laser scanning operations can be performed with a single laser. This is a working principle of commercial scanning LDVs leveraging a continuously rotating mirror [30]. However, unstable response of LDVs is frequently issued when scanning LDVs are targeted to semi or less reflective surfaces, such as liquid surface, since the mirror continuously rotates while LDVs point and shoot the laser.

In this study exclusively aiming to liquid motion measurement, an incremental rotation with pauses is adopted to operate the galvanometer scanner to enhance operating stability of LDVs as illustrated in Figure 5. A waveform of a step function drives stepwise rotation of the galvanometer scanner. A step function, with the steps equal to the number of target points to be measured, is input to the galvanometer scanner after being converted to a signal.

After the measurement is completed, the multipoint data obtained through the galvanometer scanner rotating with the step function are separated from each other at the same position to form a single point of time-series data. The constant rotation frequency of the galvanometer scanner is the measurement frequency of the separated time-series data. Due to the characteristic of the step function, spike noise is generated in addition to the data measured when moving and stopping quickly. The goal is to capture the vibration data that occurs in the stable section following the generation of the spike noise.

## 3. Experimental Investigations of the Proposed Scanning System

### 3.1. Experimental Setup

The experimental system was constructed using a tuned liquid damper (TLD) model for laser scanning, utilizing an LDV and a galvanometer scanner (Figure 6). The LDV applied to this experiment was POLYTEC’s PDV-100, which measures velocity of a target using a wavelength of 633 nm. An aluminum-coated galvanometer scanner of ThorLAB Co., Ltd., Newton, NJ, USA, GVS211, which reflects visible light in the 400–750 nm band, was applied. The lab-scale TLD model is made of a 1 cm thick acrylic plate. Its length, width, and height are 20 × 30 × 20 cm and contains 2 cm of liquid mixed with red dye. The TLD was installed on a single shaft shake table driven by an AC servo motor (HC-SFS502, MITSUBISHI). Based on the theory of shallow liquid waves, the first-order natural frequencies of the TLD are 1.09, 2.08, and 2.94Hz, respectively. In order to include all of this, the shaking table was shaken for 550 s with a chirp signal of 0.5 to 4.5 Hz. The high-definition (HD) digital camcorder (HMX-QF20, SAMSUNG) was cross-verified through a series of conventional sensors and was placed on a stationary tripod 50 cm away from the TLD. The digital camcorder is a 1920 × 1080 pixel full HD camcorder that captures 60 frames per second. The video compression format supported by the camcorder is standard MPEG compatible with MATLAB^®^.

Four points seen in Figure 7 were measured to extract the 1st–3rd order mode shapes of the TLD. In order to measure the liquid vibration of 4 points, a step function composed of 4 steps was generated. The galvanometer scanner was rotated at 12 Hz considering the Nyquist frequency for measuring the vibration signal at the maximum of 3.5 Hz. The time delay between each point is 1/48 s. The sampling frequency of the time-series data for each position is 12 Hz. After setting the LDV and the galvanometer scanner as described above, laser scanning was performed at 12 Hz from the first point to the fourth point.

### 3.2. Experimental Results

In this laser scanning experiment, some of the original data measured on the LDV are shown in Figure 8. Since the original data includes the liquid vibration data of four points to be measured and the galvanometer noise rotating at four steps of 12 Hz, it is necessary to remove the noise and extract the liquid vibration data.

The graph that enlarges the 0.2 s portion from 82.9 to 83.1 s is shown in Figure 9. When the galvanometer scanner scanned from the first point to the fourth point and then returned to the first point, the laser traveled the longest distance, resulting in the maximum noise. Based on the maximum value of the noise, the following four pieces of flat data are the four pieces of vibration data to be measured. Since the maximum value of the noise and the data of 4 points are consecutively outputted, only the data of each position is extracted from them. Then the vibration data of four points are separated. The time interval of each point is 1/48 s as input to the galvanometer scanner, and the time interval of each point after extraction is 1/12 s.

The measured data of four separate points are shown in Figure 10. While the chirp signal starts at 0.5 Hz and the frequency increases, weak sloshing occurred at 50 s at four measurement points. Weak sloshing showed the maximum value at 95 s as the size increased. The sloshing rapidly decreased after 100 s. After that, no sloshing occurred for more than 150 s. Similar to the previous sloshing, weak sloshing occurred again around 270 s. The size of the sloshing gradually increased to around 320 s, and then rapidly decreased after 320 s. Unlike the previous sloshing, weak sloshing continued after the sloshing rapidly decreased. The size of the sloshing increased around 460 s, but the size is about 20% of the size of the previous two large sloshings.

Figure 11 shows a series of graphs that compare the 4-point liquid vibration data extracted from the LDV with the video sensing data. Since video sensing can only acquire data in the out-of-plane direction, the outer wall of the TLD is measured, although the laser measures the center of the TLD. In addition, the data acquired through the video are displacement data; it was numerically differentiated and converted to velocity and compared with the velocity data of LDV. The video data showed a trend of two large sloshings and one small sloshing with the same magnitude as the LDV data for 550 s.

When sloshing occurs, waveforms of different sizes are mixed. Since the validity of the measurement of mixed waveforms is difficult to identify in the time domain, the frequency that occurs over time is derived from spectrogram analysis. In the spectrogram, frequency characteristics included in data for all time domains can be separated. The spectrogram analysis of the measured data at the first point is presented in Figure 12. Chirp signals excited from 0.5 to 4.5 Hz in all time domains are confirmed in both LDV data and video data. In the two sections where sloshing occurs, strong free surface motion generates in integer multiples of the excited frequency. As the sloshing occurs, the integer multiple frequency is repeated because the traveling wave repeatedly bounces off the wall. LDV systems utilizing galvanometer scanners can accurately measure even when multiple waveforms are mixed when sloshing occurs.

After verifying that the trends of the two data are the same, the first and fourth points in the time domain are enlarged and compared with the video sensing screen to verify the validity of the data measured by LDV. Figure 13 shows the waveform of weak sloshing that occurred from 60 s to 65 s. A waveform with multiple frequency peaks was generated, the waveform occurring at the first measurement point shifted to the fourth measurement point. This phenomenon occurs when a traveling wave on the surface propagates left and right, and the speed of this traveling wave is obtained from the movement of the peak point. The peak point of the first point that occurred at 61.7 s was acquired at the fourth point at 62.2 s, and the speed of the traveling wave is calculated to be 360 mm/s from the information of the first point and the fourth point distance of 180 mm. The video sensing image is visually confirmed, as shown in Figure 14a–c were taken at 61.7, 62, and 62.2 s, respectively. Similar to the time-series data of LDV, the wave generated from the left wall, as shown in Figure 14a, moved to the right wall as shown in Figure 14c.

As the sloshing becomes stronger over time, one peak point repeats at the first and fourth point measurements, as shown in Figure 15. The occurrence of maximum peak value and the disappearance of the sloshing immediately after the peak occurrence means that resonance has occurred. The wave’s motion comprised one large peak frequency generated after the disappearance of a small vibration shows a clear difference from the traveling wave. It is possible to estimate the resonant frequency from the repetitively occurring peak point. Since the peak of the first point that occurred at 81.8 s occurred again at 82.7 s after 0.9 s, the resonant frequency is calculated to be about 1.1 Hz; however, it was confirmed that the time sync of the sloshing motion of the first and fourth points did not match perfectly despite the resonance state because the wave was traveling at a low speed. The video sensing image was visually confirmed, as shown in Figure 16a–c were taken at 81.8, 82, and 82.2 s, respectively.

The sloshing that occurred after 270 s is similar to the sloshing that occurred after 50 s. As shown in Figure 17, the peak points of the first and fourth measurement points alternately occurred after the traveling wave was generated. The resonant frequency is calculated to be about 2.85 Hz from the peak of the first point occurring at 321.6 s and reoccurring at 321.95 s. The difference from the sloshing that occurred after 50 s is that the phase difference between the first and fourth measurement points is 180 degrees. It was confirmed that a standing wave shape appeared when a higher-order resonant frequency was excited. The video sensing image was visually confirmed, as shown in Figure 18a–c were taken at 321.6, 321.78, and 321.95 s, respectively.

### 3.3. Mode Estimation of Liquid Motion

Mode shapes were extracted by singular value decomposition with time-series data of four measurement points measured from the LDV and galvanometer scanner. The 1/48 s measurement delay of each data is disadvantageous for mode extraction because the mode is a shape that behaves simultaneously; therefore, the measurement delay is removed stochastically by taking the cross-correlation of the data for each position. Figure 19 shows the singular values of the LDV data and the video sensing data are consistent. Both data showed three peak frequencies of 1.12, 2.27, and 2.84 Hz. These peak frequencies are close as the natural frequencies 1.09, 2.08, and 2.94 Hz of the theory derived from Equation (15). It should be noted that difference between the second frequencies of 2.27 and 2.08 Hz is relatively big compared to the others, which suggests that second mode rather difficult to estimate experimentally. In fact, the second mode was not monitored in the previous sloshing examination conducted in Section 3.2.

Figure 20a–c show the mode shapes extracted from the singular vectors. The mode shapes are overall consistent with the theoretical mode shapes in Figure 2: the first and third modes are antisymmetric and the second mode is symmetric. However, lack of symmetry of the estimated second mode shape is noticed–the both end points do not match each other. The skewness of the estimated mode shape represents break of time synchronization in motion; namely the wave is still travelling wave rather than standing wave (mode). This explains the reason why the second mode did not dominantly appear during analysis of time-series data. It could be suspected that the symmetric mode is difficult to be generated by the transversal excitation adopted in this study. Based on the experimental findings, it is recommended that a careful consideration on the symmetric modes among the theoretical lower modes in the TLDs should be taken into the design stage of TLDs utilizing oscillating liquid motion to mitigate vibration of tall buildings.

## 4. Conclusions

To monitor surface motion of liquid in TLDs, a noncontact measurement method utilizing an LDV and a stepwise rotating galvanometer scanner is presented in order to overcome the disadvantages of the single measurement of the conventional contact-type wavemeter and its inability to measure the out-of-plane direction of the noncontact-type video sensor. By virtue of a stepwise rotating galvanometer scanner, a stable multipoint scanning measurement system was composed of a single point of LDV.

The validity of the proposed system is verified by comparing time-series data measured at four points of liquid vibration in the TLD with video sensing data. Further, the traveling wave and the standing wave is distinguished from the waveform of the liquid motion data. The modal analysis is successfully conducted since time delay induced by non-synchronous sensing between measurement points is stochastically removed by cross-correlation of measurement data. As a result of the singular value decomposition, the natural frequencies and mode shapes are consistent with the theoretical and conventional video sensing results.

Among sloshing surface motions of standing wave, symmetric modes are experimentally proven to be less dominant and thus attenuation of generating reaction force of TLDs against vibration of buildings should be considered at the design and operation phases of the TLDs.

## Figures and Tables

**Figure 1 sensors-21-08211-f001:**
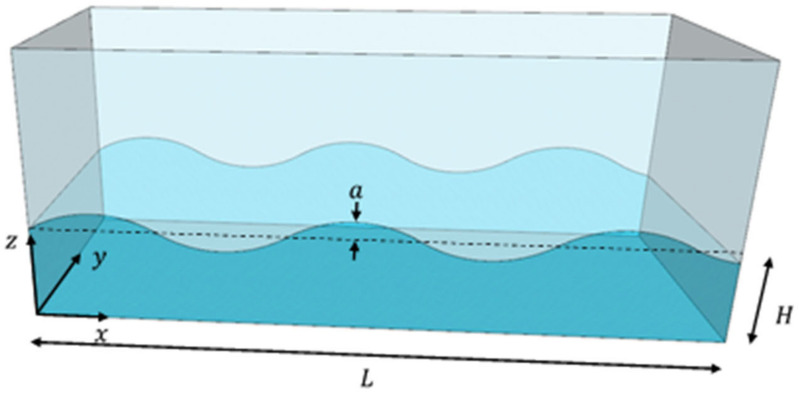
Surface wave motion of travelling wave propagation in a TLD.

**Figure 2 sensors-21-08211-f002:**
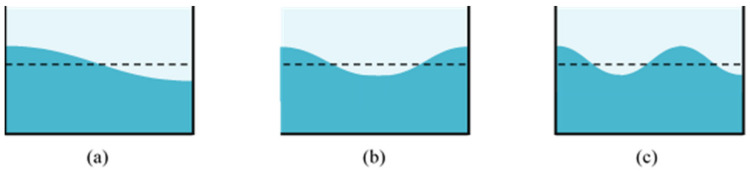
Standing wave, i.e., mode shapes, of liquid motion in a TLD: (**a**) the first frequency; (**b**) the second frequency; (**c**) the third frequency.

**Figure 3 sensors-21-08211-f003:**
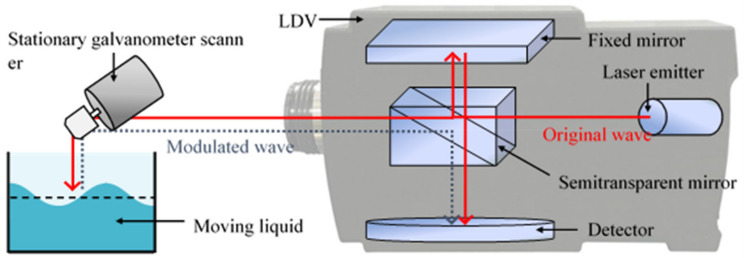
Schematic diagram of moving light in the LDV.

**Figure 4 sensors-21-08211-f004:**
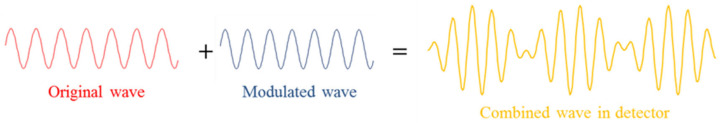
Detected laser at photodetector.

**Figure 5 sensors-21-08211-f005:**
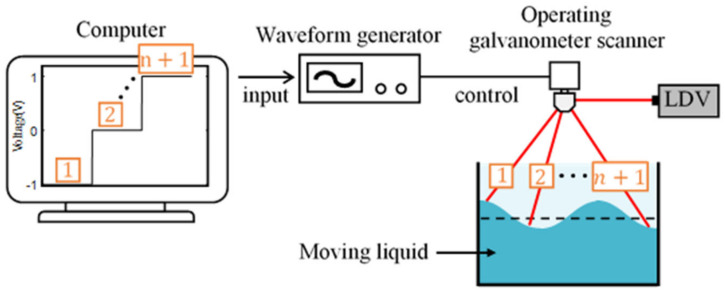
Operation of the stepwise rotating galvanometer (n: number of steps).

**Figure 6 sensors-21-08211-f006:**
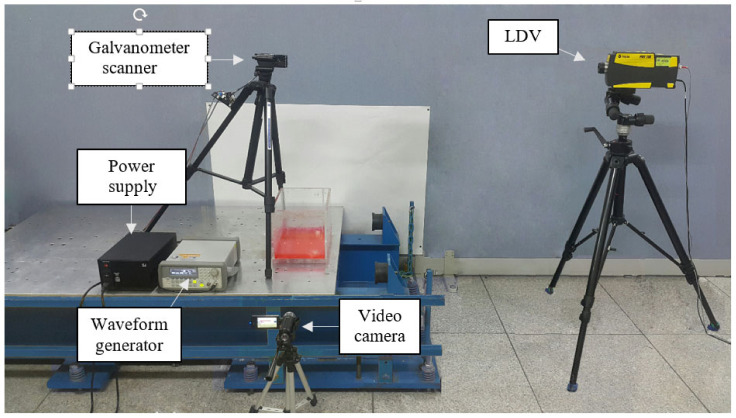
Experimental setup of the proposed wave scanning system.

**Figure 7 sensors-21-08211-f007:**
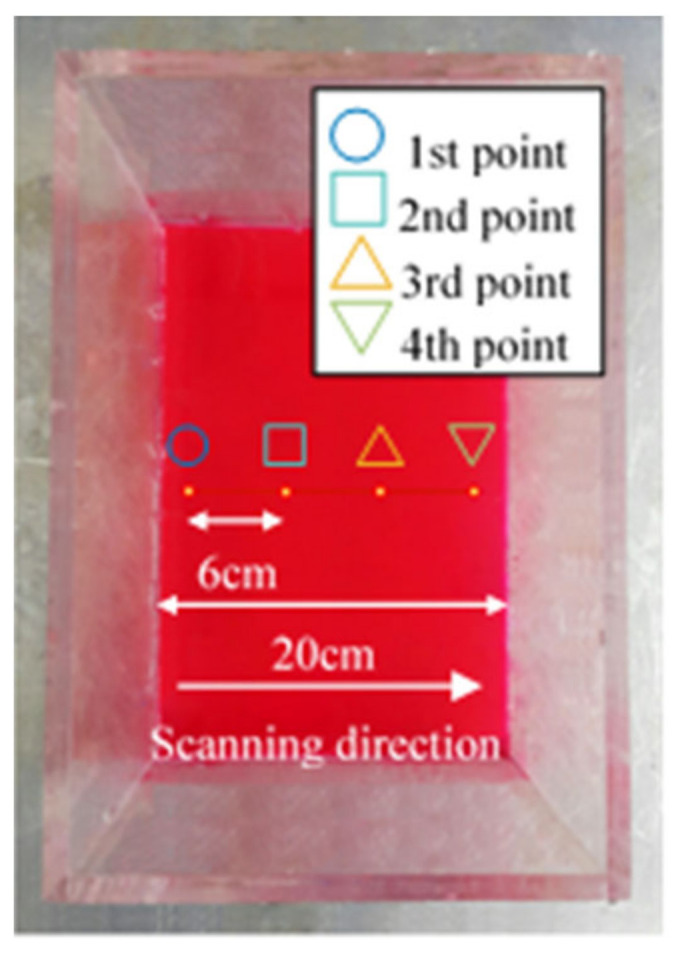
Measurement points of the rectangular tank.

**Figure 8 sensors-21-08211-f008:**
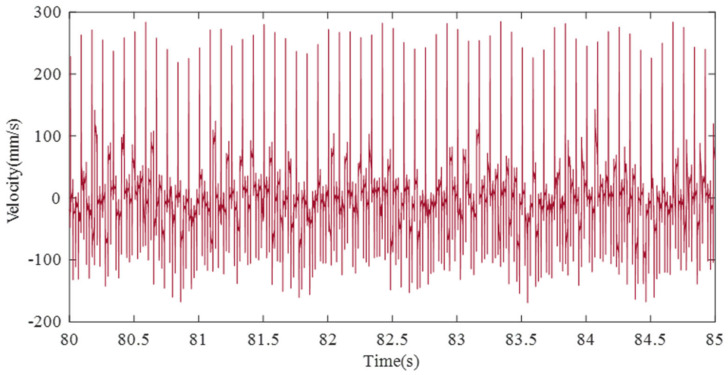
Measured data by the LDV.

**Figure 9 sensors-21-08211-f009:**
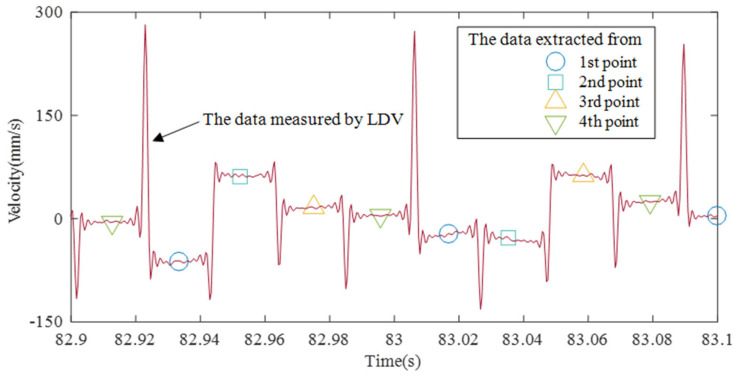
Zoomed plot of time interval of 82.9 to 83.1 s.

**Figure 10 sensors-21-08211-f010:**
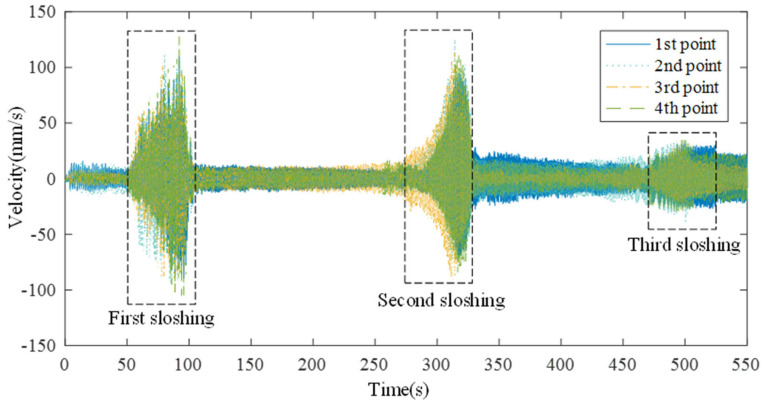
Measured data set of 4 different measurement points.

**Figure 11 sensors-21-08211-f011:**
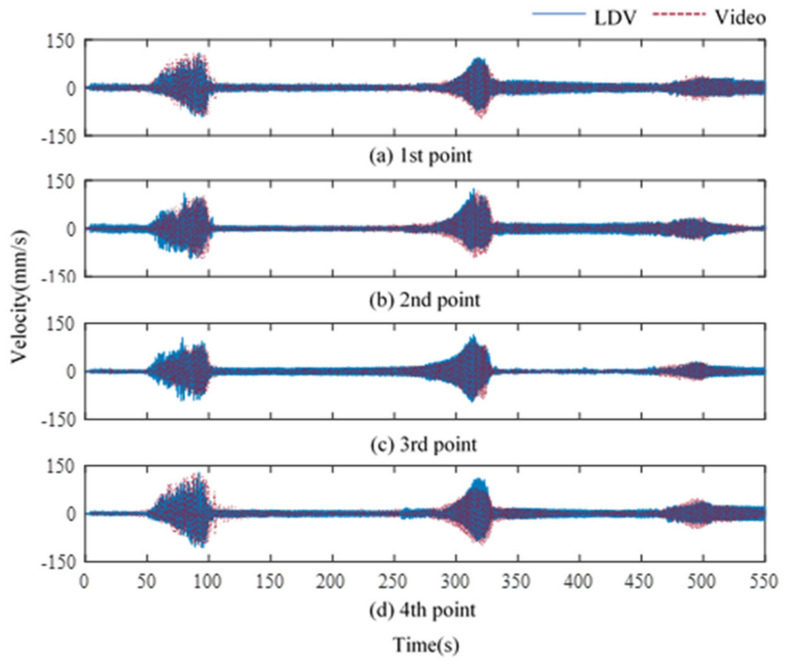
Comparison between time histories of the LDV and video camera.

**Figure 12 sensors-21-08211-f012:**
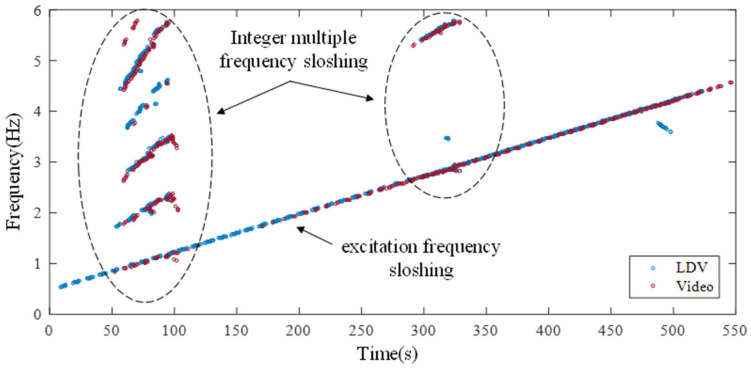
Spectrogram analysis of the first point over time.

**Figure 13 sensors-21-08211-f013:**
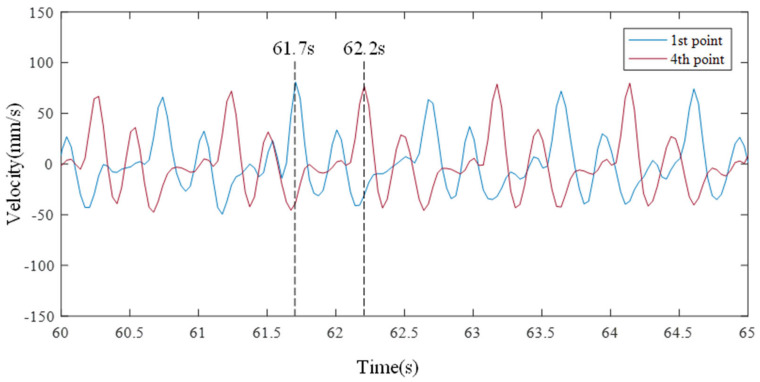
Propagation waves occur due to vibrations close to natural frequencies.

**Figure 14 sensors-21-08211-f014:**
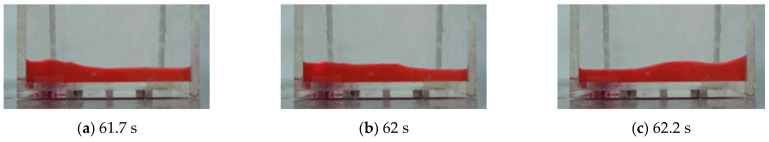
Still shots of wave propagation wave recorded by the video camera.

**Figure 15 sensors-21-08211-f015:**
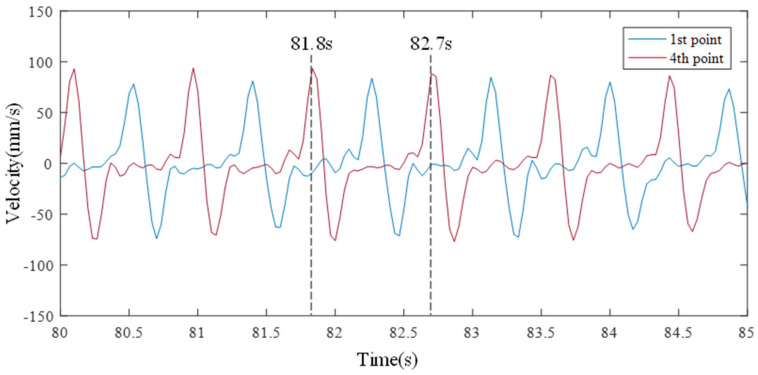
Time histories at the first natural frequency.

**Figure 16 sensors-21-08211-f016:**
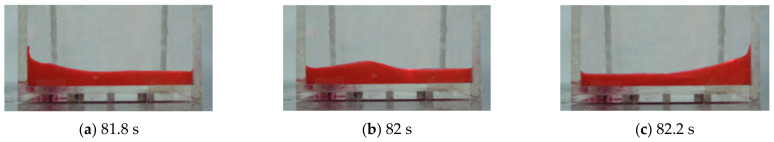
Still shots of a first natural frequency wave recorded by the video camera.

**Figure 17 sensors-21-08211-f017:**
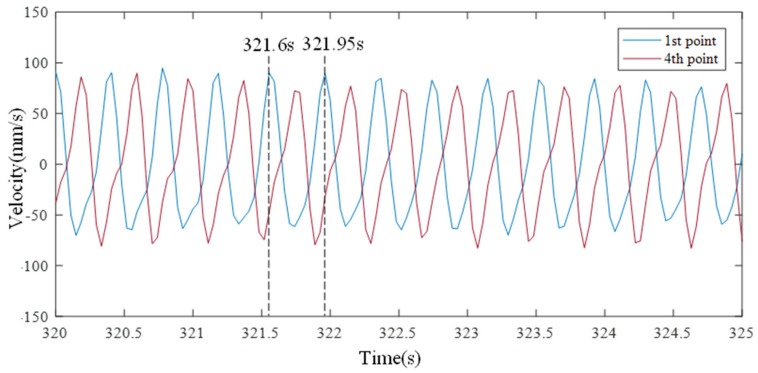
Time histories at the third natural frequency.

**Figure 18 sensors-21-08211-f018:**
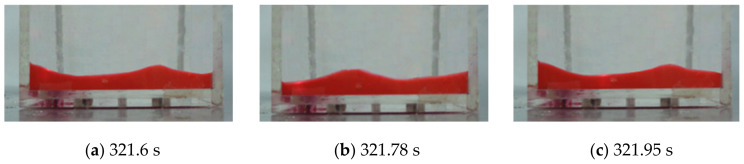
Still shots of a third natural frequency wave recorded by the video camera.

**Figure 19 sensors-21-08211-f019:**
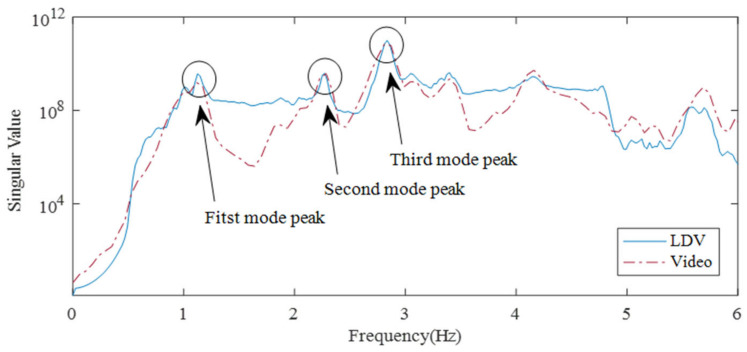
Singular value plot for standing wave extraction.

**Figure 20 sensors-21-08211-f020:**
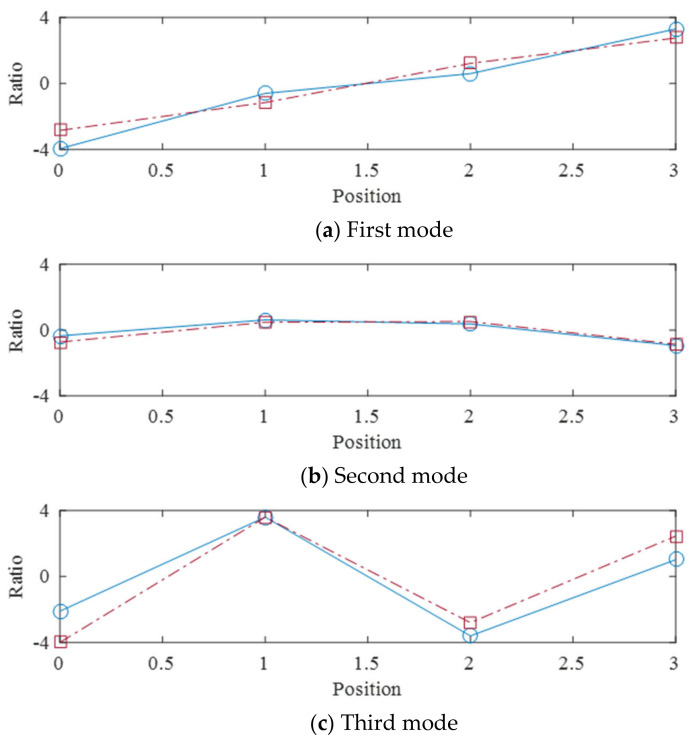
Mode shapes extracted from singular value decomposition (solid line: estimated, dashed line: theory).

## Data Availability

Not applicable.

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
