# Peer review of "Multipoint Wave Measurement in Tuned Liquid Damper Using Laser Doppler Vibrometer and Stepwise Rotating Galvanometer Scanner"

_sensors, 2021, doi:10.3390/s21248211_

Round 1
Reviewer 1 Report
See attached document.

Author Response
See attached document.

Reviewer 2 Report
- Section 2.1 is about the sloshing frequencies of a rectangular basin that has appeared in the textbook. If the aim of this study is not the dynamic modeling, this section needs to be shortened.
 - Section 2.2 is about the working principle of the LDV but there is no reference for this part. If Figure 5 is a typical diagram of the LDV measurement, then this part might be omitted.
 - Section 2.3 is about the proposed scanning system. However, it is not clear whether this system was originally developed by authors. Has the combination of the galvanometer and LDV not been attempted by other researchers?
 - Experimental sloshing frequencies are slightly different from theoretical predictions. What is the source of error?
 - Authors talked about nonlinear phenomena but didn't explain it.
 - If the objective of this study is only to identify the sloshing frequencies experimentally, then what is the advantage of the proposed method over other approaches?

Author Response
See attached document.
